# CRISPR/Cas9 Genome-Editing Technology and Potential Clinical Application in Gastric Cancer

**DOI:** 10.3390/genes13112029

**Published:** 2022-11-04

**Authors:** Renata Sanches Almeida, Fernanda Wisnieski, Bruno Takao Real Karia, Marilia Arruda Cardoso Smith

**Affiliations:** 1Discipline of Genetics, Department of Morphology and Genetics, Federal University of São Paulo, Rua Botucatu, 740, São Paulo 04023900, Brazil; 2Discipline of Gastroenterology, Department of Medicine, Federal University of São Paulo, Rua Loefgreen, 1726, São Paulo 04040002, Brazil

**Keywords:** gastric cancer, CRISPR, gene editing

## Abstract

Gastric cancer is the subject of clinical and basic studies due to its high incidence and mortality rates worldwide. Due to the diagnosis occurring in advanced stages and the classic treatment methodologies such as gastrectomy and chemotherapy, they are extremely aggressive and limit the quality of life of these patients. CRISPR/Cas9 is a tool that allows gene editing and has been used to explore the functions of genes related to gastric cancer, in addition to being used in the treatment of this neoplasm, greatly increasing our understanding of cancer genomics. In this mini-review, we seek the current status of the CRISPR/Cas9 gene-editing technology in gastric cancer research and clinical research.

## 1. Introduction

Gastric cancer (GC) remains an important cancer worldwide, ranking fifth for incidence and fourth for mortality globally [1]. The high mortality rates of GC are mainly due to its detection at advanced stages, where it can no longer be treated with curative intent, and to the few available therapeutic options [2]. Currently, gastrectomy is the main option for locoregional GC treatment [3]. However, even after curative resection and adjuvant chemotherapy, the clinical outcome remains poor, with a recurrence rate of approximately 35% [4]. Despite some recent advances that have been made in the treatment of GC (e.g., systemic chemotherapy, radiotherapy, surgery, immunotherapy, and targeted therapy), only a subset of patients can benefit from these treatment options [5].

GC is a complex, heterogeneous, and multistep disease that involves environmental factors, mainly *Helicobacter pylori* infection, and several molecular alterations, such as genetic instability, the inactivation of tumor suppressor genes, and the activation of oncogenes.

Moreover, the molecular profiles may vary from patient to patient. Although many studies proposed a great number of diagnostic and prognostic markers, only a few of them are currently used in clinical practice [6]. Furthermore, different epigenetic alterations have been reported to activate oncogenes and inactivate tumor suppressor genes during gastric carcinogenesis [7]. Therefore, an understanding of the molecular aspects involved in gastric carcinogenesis and tumor heterogeneity is essential to providing new multidisciplinary treatments [8].

In this context, gene-editing technology has received increasing attention in GC, as it can adjust gene expression alterations and correct mutations. In addition, this technology can also reveal the roles of unknown genes in important pathways, and consequently can help in the identification of new biomarkers and therapeutic targets, as well as the mechanisms of GC responses to drug treatment [9].

Since Kim et. al.’s study, the methodologies for gene editing have involved the use of the zinc finger nuclease (ZFN) and the transcription activator effector nuclease (TALENs) by targeting DNA domain-binding proteins [10,11,12,13,14]. Currently, clustered regularly interspaced short palindromic repeat (CRISPR)-associated 9 is a gene-editing tool with a lower cost, higher efficiency, and less complexity in its application. In addition, the 2020 Nobel Prize in Chemistry was awarded to Emmanuelle Charpentier and Jennifer Doudna for the development of the CRISPR/Cas9 gene-editing technology [15].

In this mini-review, we first describe the mechanism and development of the CRISPR/Cas9 gene-editing system. Furthermore, we focus on the current applications of this technique for the basic research, diagnosis, and therapy of GC. The potential applications of CRISPR/Cas9 in GC therapy and the challenges are discussed.

## 2. The CRISPR/CAS9 Technology

The first hint of CRISPR was in 1987, when Japanese scientists studied the alkaline phosphatase gene in *Escherichia coli* and discovered an adjacent region of unknown function [16]. After two decades, it was discovered that these previously unknown regions were related to the adaptive immune systems of bacteria and prokaryotic organisms such as archaea [17,18,19]. This finding favored the discovery of the association of the catalytic enzyme Cas9 with the CRISPR system, providing the double-stranded DNA breaks [20]. In 2013, the CRISPR/Cas9 system was used for the first time as a gene-editing tool [21,22,23].

CRISPR/Cas9 works with a simplified system using a guide RNA (gRNA) that can recognize the genomic target through base complementarity according to the Watson–Crick theory, and also has the function of aggregating the endonuclease Cas9. Due to its simple system, greater specificity, and efficiency, the CRISPR/Cas9 system is a much more effective tool than the previous ZFN and TALENs methodologies [24].

The CRISPR methodology has been improved over time (Figure 1), and to determine the specificity of the machinery connection, a PAM (protospacer-adjacent motif) primer sequence of 3 nucleotides (NGG) before the 20 nucleotides of the target is determined. In addition, the Cas type II protein from *Streptococcus pyogenes* (SpCas9) has been used for the break of the double strand of DNA. In response to DNA cleavage, the host cell can perform two different repair mechanisms. The non-homologous end joining (NHEJ) is a mechanism where the ends of the tape come together, which is subject to unwanted insertions or deletions. Another mechanism is homology-directed repair (HDR), which uses recombinations already determined from DNA models to reconstitute the cleaved DNA (Figure 2).

In addition, large chromosomal deletions, inversions, and translocations [33], as well as plasmid and retrotransposon insertions [34,35], have also been reported in the literature as a result of the DSBs induced by CRISPR/Cas9.

Accordingly, the ability of CRISPR/Cas9 to generate DNA double-stranded breaks at sites of interest, introducing a donor DNA template into the target cell, means it has been proven to be a useful tool for genome editing [36,37].

Moreover, using a modified version of the Cas9 protein, dCas9 (catalytic “dead” Cas9), it is possible to target desirable regions on the genome without cutting the DNA strands [25,38], which enables the possibility of adding many different proteins to dCas9, including fluorescent proteins. In 2013, Chen and collaborators described a method for the imaging of repetitive elements in telomeres and coding genes in living cells using a CRISPR/dCas9-EGFP tag [26]. Interestingly, regulatory factors such as modulators of gene expression can also be fused to dCas9. Like RNAi (RNA interference), CRISPRi (CRISPR interference) may be used to repress gene expression, although using a different mechanism. While RNAi uses a nucleotide base complementary to the desired sequence and the RNA-induced silencing complex (RISC) to suppress an mRNA target [39], CRISPRi uses a dCas9 linked to a transcription repressor domain to identify and bind to a specific DNA locus, and then to inhibit its transcription [40]. On the other hand, it is also possible to conjugate a dCas9 to a transcription activator and to create a CRISPR activation (CRISPRa) system [40]. Hilton and colleagues revealed that CRISPR/dCas9 fused to the human acetyltransferase p300 can be used to activate gene transcription by acetylating histone H3 lysine 27 at its DNA target sites (Figure 3) [27].

Additionally, a dCas9 may be engineered to introduce point mutations to a specific DNA sequence without causing DSBs. Fusing a single-stranded DNA deaminase enzyme to dCas9 allows for single-nucleotide modifications at a genomic sequence of interest. Taking advantage of a segment of accessible single-stranded DNA formed by dCas9, gRNA, and target DNA, so far cytidine and adenine deaminases have been used to convert cytosine into uracil and adenosine into inosine, respectively [28,29,41].

In the context of gastric diseases, Krishnamurthy and collaborators studied gastric organoids derived from two individuals with homozygous mutations in PDX1 (pancreatic and duodenal homeobox 1), causing a loss of its expression. PDX1188delC/188delC organoids resulted in gastritis, a loss of antral identity, and gastric and intestinal metaplasia. Interestingly, the restoration of PDX1 protein expression via PDX1 point mutations using CRISPR/Cas9 also reversed these phenotypes. These results suggest that the CRISPR-mediated correction of point mutations has the potential to improve patient care in the future [42].

Garcia-Bloj et al. [43], by comparing the ZFN, TALENs, and CRISPR technologies for the overexpression of tumor suppressors in GC strains, demonstrated that CRISPR-dCas9 can restore the expression of the *MASPIN/REPRIMO* genes, being considered a transient, targeted, and efficient tool for restoring tumor suppressor functions.

## 3. Application of CRISPR/CAS9 in Basic GC Research

CRISPR/CAs9 is an extremely versatile tool that can be used in the study and understanding of cancer development mechanisms and in the diagnosis and treatment of several diseases, including GC [15].

This technique has helped studies in several areas, such as for gastric cancer, where in 2015 the study by Gannom and colleagues used the CRISPR/Cas9 technique for the first time in GC [44]. This pioneering study evaluated the knockdown effect of *dual-specificity mitogen-activated protein kinase 1* (*MAP2K1*) and the relationship of MEK-inhibitory drugs with cancer cell lines, including gastric cancer. This study was important to evidence the RAS/MAPK activation driven by *MAP2K1* depletion in gastric cancer.

CRISPR is an important tool that [9] helps us in a simple and practical way to understand the functions of genes that are still poorly described in GC. A collection of relevant GC studies that used CRISPR/Cas9 may be found in Table 1.

### 3.1. Cell Viability and Proliferation

Needless to say, the ability of cells to proliferate rapidly and uncontrollably is a key factor in GC. Therefore, understanding these biological mechanisms and genes is extremely important for the identification of therapeutic targets. The CRISPR/Cas9-mediated knockout of *apoptosis-associated speck-like protein containing a C-terminal caspase recruitment domain* (ASC) blocked IL18 and augmented apoptosis in human GC cells, helping to reveal a novel pro-tumorigenic ASC/IL18 signaling axis in GC cell survival and a candidate therapeutic target in this disease [55].

Contributing to the findings of mir-21-induced loss of 15-hydroxyprostaglandin dehydrogenase (15-PGDH) in early GC and its phenotypic consequences, another study suppressed *15-PGDH* in GC cells using CRISPR/Cas9, which demonstrated increased GC cell proliferation [51]. Therefore, the study suggested that maintaining the 15-PGDH enzyme activity could be a strategy for preventing GC, especially in tubular adenocarcinoma.

### 3.2. Cell Cycle Control

The study of genes that influence the cell cycle allows us to identify important cellular processes that differentiate a tumor cell from a healthy cell. Aspects such as unrestrained growth without the influence of external factors, the loss of programmed cell death capacity, and also the repression of tumor suppressor genes, among other properties, are important to identify specific biomarkers and potential therapeutic targets in GC.

To demonstrate the involvement of DNA hypermethylation on the regulation of the tumor suppressor gene *REPRIMO* and the *TP53-dependent G2 arrest mediator homolog* (*RPRM*) in GC, Lai and collaborators [58] used the CRISPR/Cas9 technology to first knockdown DNA methyltransferases (DNMTs). The knockdown of *DNMT3A* and *DNMT3B* resulted in a significant increase in *RPRM* mRNA by decreasing the *RPRM* promoter methylation in GC cells, suggesting an inverse correlation between *DNMT* functions and *RPRM* gene expression. To confirm the tumor suppression role of *RPRM*, the authors generated RPRM-deficient GC cell lines using CRISPR/Cas9, which were further inoculated in mice. The loss of *RPRM* enhanced the tumor formation in the in vivo model, confirming the role of *RPRM* as a tumor suppressor gene in GC. This study provided information regarding the role of *RPRM* and its regulatory methylation mechanism in GC development with potential application as a therapeutic target.

To regain the tumor suppression gene function, a study reactivated the *RPRM* expression using CRISPR/dCas9 linked to VP64 and SAM effector domains in GC cell lines, showing a hypermethylated *RPMR* promoter and expressing very low basal levels of the gene [43]. As a result, a marked reduction in GC proliferation was observed. This result outlined the advantage of this combinatorial epigenome editing approach to reactivate highly methylated tumor suppressor genes as a promising therapy for GC.

Another promising tumor suppressor was described in the study by Hu and collaborators [52], with the knockout of the *Morf4-family-associated protein 1* (*MRFAP1*) gene using CRISPR/Cas9 from the CG human cell lines. The researchers observed that the knockout promoted an interaction of this *MLN4924* subtract with suppressed tumor proteins, such as p27, which promoted a decrease in vitality, increased cell cycle arrest, and apoptosis, data that are important to relate the gene–subtract interaction to favor improvements in cancer treatment. The study by Liu et al. [56] used the CRISPR/Cas9-mediated knockout of *promoter of CDKN1A antisense DNA damage-activated RNA* (*PANDAR*) and evaluated its effects on the phenotype of GC cell lines. The knockout of *PANDAR* suppressed the proliferative activity and colony formation in the GC cell line. It was also observed by flow cytometry that the *PANDAR* knockout blocks the progression of the cell cycle at the G1/S checkpoint. Through this unique pattern of transcriptional modification, *PANDAR* remarkably facilitated the proliferation of cancer cells, the formation of clones, and resistance to chemotherapy. Another study developed by Zhang and colleagues [53] used the CRISPR/Cas9-mediated knockout of *LIM homeobox transcription factor 1 α* (*LMX1A*) in GC cell lines to identify *LMX1A* as a primary target of miR-9. The authors that demonstrated the knockout of *LMX1A* increased the cell viability and cell proliferation, reinforcing the role of *LMX1A* as a tumor suppressor in GC.

### 3.3. Invasion and Migration

The basic mechanism of metastasis development involves several important characteristics, such as the ability of cells to detach themselves from the primary tumor site and migrate to colonize more distant sites in the organism [64]. Due to the lack of early diagnosis and effective therapy options for GC, a better understanding of the mechanisms involved in the metastatic process of GC is necessary. In this sense, Zhou and colleagues [49] used the CRISPR/Cas9-mediated knockout of *gastric cancer metastasis-associated long non-coding RNA* (*GMAN*) IncRNA and evaluated the phenotype of GC cells. The knockout of the *GMAN* lncRNA delayed the invasive activity of these cells. This study also showed the overlapping relationship between *GMAN* and *ephrin A1*, in which the *GMAN* knockout induced an *ephrin A1* expression reduction. Assessing the effect of *ephrin A1* knockout in GC cell lines, it reduced the ability for invasion and metastasis in in vivo experiments. Another important study was developed by Zhu and colleagues [60], in which a *GWAS* analysis was used to find low-frequency genetic variants associated with the risk of GC. The authors found two genes that contained a variant associated with GC risk, *SPOC-domain-containing 1* (*SPOCD1*) and *Butyrophilin subfamily 3 member A2* (*BTN3A*), and eliminated these genes using CRISPR/Cas9 in GC cell lines. The knockout of *SPOCD1* and *BTN3A2* inhibited cell proliferation and colony formation. To investigate the effects of *SPOCD1* and *BTN3A2* knockout on migration and invasion, the authors performed xenograft assays and observed a tumor growth reduction in a rat model associated with *SPOCD1* knockout. On the other hand, *BTN3A2* was suggested as a susceptibility gene, although no significant changes in the xenograft model were observed. Zhang et al. [48] demonstrated that the CRISPR/Cas9-mediated deletion of *SAM-pointed domain-containing Ets transcription factor* (*PDEF*) inhibited the apoptosis, colony formation, migration, and invasion of GC cell lines. These results confirm the involvement of *PDEF* in the different stages of GC development. The study by Araújo et al. [50] demonstrated that the CRISPR/Cas9-mediated elimination of the *Piwi-like protein 1* (*PIWIL1*) gene decreased GC cell migration and invasion abilities, demonstrating the oncogenic role of *PIWIL1*. On the other hand, Chen Wei’s study [63] demonstrated that the CRISPR/Ca9-mediated knockout of *Somatostatin* (*SST*) significantly promoted the migration and invasion capabilities of GC cell lines. These data are essential for characterizing *SST* as a potential tumor suppressor in GC.

Another interesting study used the CRISPR/Cas9-mediated knockin of *ephrin type-B receptor 2* (*EphB2*) and evaluated its functions as an independent prognostic marker in patients with GC [62]. The results indicated that the activation of *EphB2* in GC cells increased the malignant properties of GC cells, reducing the adhesion but accelerating the migration and invasion capabilities. These results indicate that *EphB2* plays a pro-tumor role in GC and has therapeutic potential to be used in this neoplasia.

### 3.4. Tumorigenesis Models

*Helicobacter pylori* is a Gram-negative spiral bacterium that is present in 58% of the global population [65]. Most individuals infected with *H. pylori* are asymptomatic, and the presence of this bacterium increases the risk of developing ulcers and gastric adenocarcinoma [66]. An interesting study was carried out by Hu and collaborators [49], who sought to demonstrate the mechanisms in which vitamin D3 can assist in the defense of the host by promoting an autophagic reaction in the fight against *H. pylori*. The results showed that there is a new pathogenic mechanism that *H. pylori* can survive by hiding inside the autophagosomes in the GC cells by using the CRISPR/Cas9-mediated knockout of *protein disulfide-isomerase A3* (*PDIA3*). From this study, a new vitamin D3 signaling pathway activates the PDIA3-STAT3-MCOLN3-Ca 2+ axis to reactivate the lysosome.

The molecular mechanism by which *H. pylori* induces peptic ulcers or gastritis cancer is not understood, but it probably involves a combination of host genetic predisposition and bacterial virulence factors (e.g., VacA and CagA proteins) [65]. The vacuolating cytotoxin (*VacA*) is responsible for several cellular responses, such as cell vacuolization, as well as other processes such as autophagy and necrosis [67]. Foegeding and collaborators [47] inhibited autophagy by using the CRISPR/Cas9-mediated knockout of *autophagy-related 16-like 1* (*ATG16L1*) in HeLa cells. The results showed increased *VacA* levels or increased vacuolization compared with the control and that the *VacA* degradation is independent of the autophagic activity. *CagA* is a virulence factor used for the detection of *H. pylori* and is considered an important risk factor for severe gastric diseases, including GC. Zhao and coauthors [54] evaluated the effects of integrin receptors and the function of *cell adhesion molecule 1 receptors* (CEACAM) through the cag-type IV secretion system (cag-T4SS) on the CagA translocation process through a multiple knockout of CEACAM receptors in the GC cell line. The results showed that neither the direct interaction of the components of cag-T4SS with the integrins nor any signaling event mediated by the integrin is necessary for the translocation of *CagA*. Furthermore, the CRISPR/Cas9 mediated the deletion of miR-30a in *H. pylori* infected mice, the knockout mice demonstrated that genetic editing did not affect the growth and development of the mice, and little effect was observed on the *H. pylori* colonization rates of the mice. Increased incidence rates of chronic gastritis, chronic atrophic gastritis, atypical hyperplasia, and other precancerous lesions and manifestations of adenocarcinoma in the antral or gastric mucosa of rats have also been reported. These data demonstrate that miR-30a plays the role of a tumor suppressor in GC [59].

### 3.5. Chemotherapy Response

From the development of tumor cells to create resistance to chemotherapeutic-induced cell death, the CRISPR tool has been used to discover new therapeutic targets and drugs for the treatment of GC. Wang and colleagues [57] used CRISPR/Cas9 to mediate the knockout of the *gasdermin E* (*GSDME*) gene in a GC cell line to assess the effect of 5-fluoracil on the induction of pyroptosis in these cells. The authors found that the *GSDME* deficiency changed the pyroptosis induced by 5-FU to apoptosis, characterized by shrinkage, fragmentation in apoptotic bodies, and cell death without lysis. The study by Cui and collaborators [61] used the CRISPR/Cas9-mediated knockout of the *legumain* (*AEP*) gene to assess the proliferative capacity of these cells in the presence of different chemotherapeutic agents. This gene has previously been shown to be an oncogene related to invasiveness and metastasis in GC. The authors demonstrated that *AEP* knockout GC cells caused significantly decreased proliferation after treatment with 5-FU, paclitaxel, docetaxel, and T-DM1. These data demonstrate that *AEP* is a potential therapeutic target for GC.

## 4. Application of CRISPR/CAS9 in GC Clinical Research

Due to the ease of use that CRISPR/Cas9 has brought to gene editing, the chances of treating diseases previously defined as “incurable” has greatly increased. However, this technology is a challenge in the field of clinical studies related to GC, due to the difficulties in finding a delivery system that is specific enough and capable of targeting only the cells of interest inside a given tissue.

There are currently two clinical trials where CRISPR/Cas9 is being used for the treatment of patients with GC via different approaches. The “Phase I/II Trial in Patients with Metastatic Gastrointestinal Epithelial Cancer Administering Tumor-Infiltrating Lymphocytes in which the Gene Encoding CISH was Inactivated using the CRISPR/Cas9 System” (NCT04426669) is one of the pioneers in this type of study, which is a clinical trial to evaluate the safety and efficacy of genetically modified neoantigen-specific tumor-infiltrating lymphocytes (TILs), in which the intracellular immune checkpoint CISH (cytokine-induced SH2 protein) will be inhibited using CRISPR/Cas9 gene editing for the treatment of gastrointestinal (GI) cancer [68].

Another interesting ongoing clinical trial is “PD-1 Knockout EBV-CTLs for Advanced-Stage Epstein–Barr Virus (EBV)-Associated Malignancies”. This clinical trial seeks to collect peripheral blood lymphocytes from patients with advanced stage Epstein–Barr virus (EBV) GC to generate PD-1 (programmed cell death protein 1) knockout cytotoxic T-lymphocytes (CTL). After the reintroduction of PD-1 knockout EBV-CTL, the patients are expected to show increased GC progression-free survival (PFS) and overall survival (OS) rates [69].

These studies are important for understanding the administration of CRISPR in humans and establishing this technique as a future personalized treatment in patients with GC.

## 5. Conclusions and Future Perspectives

Over the years, CRISPR technology has given us new perspectives that were previously unattainable in the scientific world. In the context of GC, CRISPR has been an ally in the discovery of new mechanisms and genes that play key roles in this neoplasm. In addition to helping to understand the molecular mechanisms involved in the emergence of GC, CRISPR is a promising tool with the potential to modify specific genes important for gastric carcinogenesis and reverse processes as we have never been able to before. CRISPR technology has proved to be such an advantageous tool, as we can see its potential via in vitro, in vivo, and clinical studies, proving it to be a technology that is applicable for several approaches, including in cancer, but we still have not identified the complete domain of CRISPR, so there is still a lot of work to be done to refine this technology and make it applicable.

## Figures and Tables

**Figure 1 genes-13-02029-f001:**
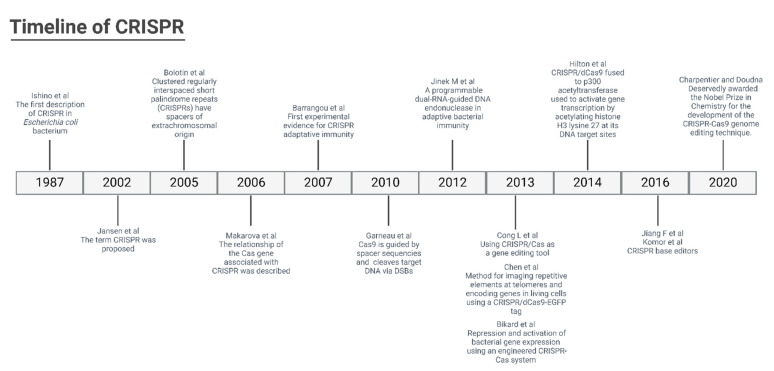
Timeline of CRISPR technology [16,17,18,19,21,25,26,27,28,29,30,31,32].

**Figure 2 genes-13-02029-f002:**
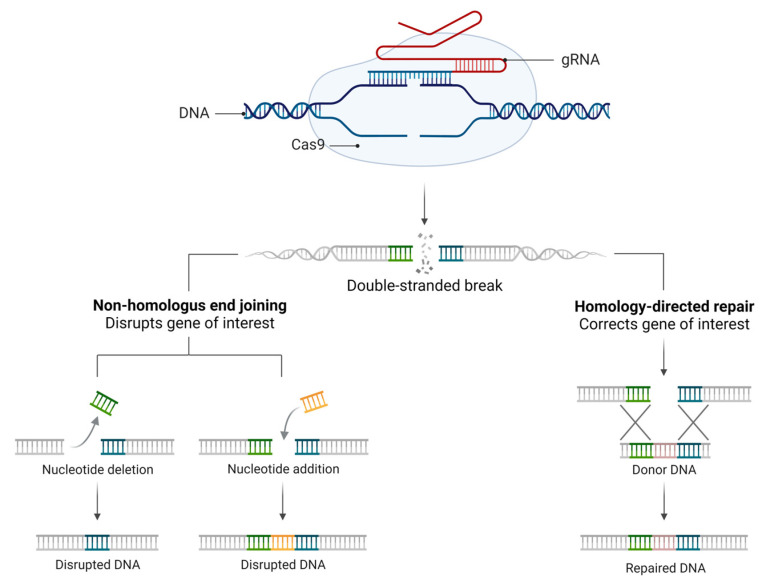
CRISPR repair mechanism. Classic CRISPR mechanism comprising the guide RNA (gRNA), target DNA, and endonuclease (Cas9), which promotes DNA cleavage by DSBs induced by Cas9. DSBs can be repaired by two main mechanisms: the NHEJ mechanism, where the ends of the strands come together (this is subject to unwanted insertions or deletions), or by the HDR pathway, which uses the recombination donor DNA templates to reconstitute the DSB.

**Figure 3 genes-13-02029-f003:**
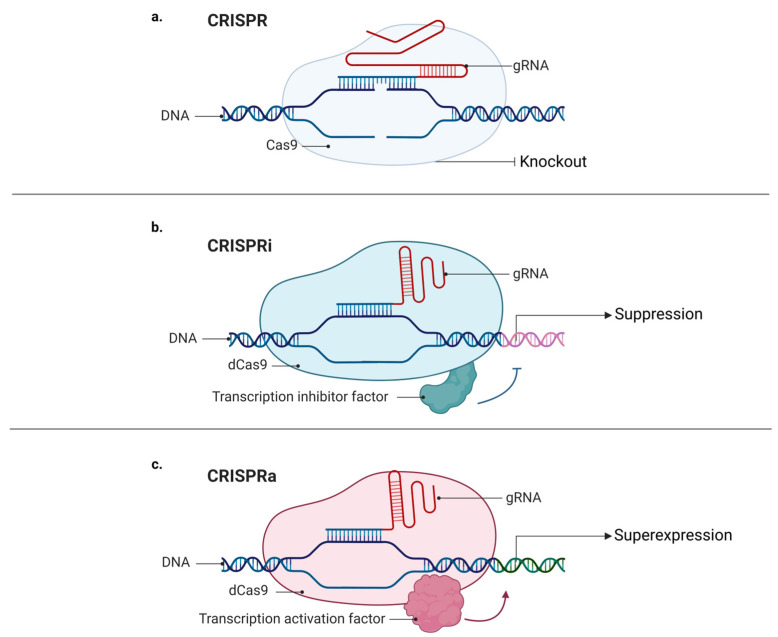
Mechanisms of CRISPR, CRISPRa, and CRISPRi: (**a**) classic CRISPR mechanism comprising a guide RNA (gRNA), target DNA, and endonuclease (Cas9), which promotes DNA cleavage, resulting in a knockout; (**b**) CRISPR mechanism comprising a guide RNA (gRNA), sequence target (target DNA), and “dead” endonuclease (dCas9), without its catalytic domain, fused with inhibitory transcription factors favoring decreased expression and generating knockdown; (**c**) CRISPR mechanism comprising a guide RNA (gRNA), target sequence (target DNA), and “dead” endonuclease (dCas9), without its catalytic domain, fused with transcriptional activating factors, favoring increased expression and causing upregulation of a given gene.

**Table 1 genes-13-02029-t001:** Studies performed in gastric cells using the CRISPR/Cas9 methodology.

Gene	Cell Line	CRISPR Approach	Phenotype	Gene Classification	Functional Analyzes Performed	Reference
*FGFR*	KATOIII, SNU16, AGS	Knockout	Cell proliferation, migration, differentiation, and cell death	Oncogene	Cell viability	[45]
*PDIA3*	HFE145 e GES-1	Knockout	Stress-resonant protein.	NE	Colony formation	[46]
*ATG16L1*	AGS, HeLa	Knockout	Autophagy	NE	Cell viability	[47]
*PDEF*	GES, SGC, AGS	Knockout	Transcription factor	Oncogene	Proliferation, apoptosis, colony formation, migration, and invasion	[48]
*GMAN*	BGC-823, SGC-7901 e MKN45, GES-1 HGC-27, MGC-803, AGS	Knockout	Metastasis.	Oncogene	Invasion assay, Cell cycle, proliferation, and colony formation	[49]
*PIWIL1*	AGP01	Knockout	Cell proliferation	Oncogene	Proliferation, apoptosis, colony formation, migration, and invasion	[50]
*MicroRNA-21*	TMK-1, AGS, KATO III, NCI-N87, MKN-1, MKN-28, MKN-45, SNU-1, SNU-5, SNU-216, SNU- 484, SNU-601, SNU-638, SNU-668 e SNU-719	Knockout	Regulation of prostaglandins in carcinogenesis	Oncogene	Proliferation, apoptosis, colony formation	[51]
*MRFAP1*	AGS, SGC-7901	Knockout	Cell cycle	Tumor suppressor	Cell cycle, cell viability	[52]
*LMX1A*	AGS primary cells C-1/GC-2	Knockout	Transcription factor	Tumor suppressor	Viability, colony formation, TUNEL (cell death)	[53]
*CEACAM*	AGS, KatoIII	Knockout	Cell adhesion related to the carcinoembryonic antigen	NE	NE	[54]
*ASC*	AGS, MKN1	Knockout	Proinflammatory cytokine	Oncogene	Apoptosis, colony formation assay	[55]
*PANDAR*	AGS SNU-1	Knockout	LNC RNA promoter of RNA activated by damage to antisense DNA CDKN1A	Oncogene	Cell viability, proliferation, and colony formation	[56]
*GSDME*	SGC-7901, MKN-45, HL-60	Knockout	Cell death by pyroptosis	Tumor suppressor	Cell Viability, LDH Release Assay, Cell Death, and Apoptosis	[57]
*REPRIMO*	BGC-823, AGS GES-1	Knockout	Cell cycle	Tumor suppressor	MTT	[58]
*miR-30a*	MKN45 SGC-7901 HEK293T	Knockout	Post-transcriptional regulation	Tumor suppressor	Proliferation, migration, colony formation, viability	[59]
*SPOCD1*	BGC823, HGC27, MGC803, SGC7901, MKN28 GES1	Knockout	Promotes migration and apoptosis reduction in CG	Oncogene	Proliferation, colony formation, migration, and invasion	[60]
*BTN3A2*	BGC823, HGC27, MGC803, SGC7901, MKN28 GES1	Knockout	Adaptive immune response	Oncogene	Proliferation, colony formation, migration, and invasion	[60]
*AEP*	HEK293T, MKN45 e SGC7901	Knockout	Lysosomal protein	Oncogene	Cell viability	[61]
*MASPIN*	HEK293T, MCF7, SUM159, H157	Knockin	Mammary protease serine inhibitor	Tumor suppressor	Cell viability, apoptosis, and proliferation	[43]
*REPRIMO*	HEK293T, MCF7, SUM159, H157	Knockin	Cell cycle	Tumor suppressor	Cell viability, apoptosis, and proliferation	[43]
*EphB2*	HGC27	Knockin	Migration and invasion	Oncogene	Cell viability, apoptosis, and proliferation	[62]
*SST*	293T and BGC823	Knockout	Migration and invasion	Tumor suppressor	Cell viability, apoptosis, and proliferation	[63]

NE: not evaluated.

## Data Availability

Not applicable.

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
