# Peer review of "CRISPR/Cas9 Genome-Editing Technology and Potential Clinical Application in Gastric Cancer"

_genes, 2022, doi:10.3390/genes13112029_

Round 1
Reviewer 1 Report
11. In the section of “The CRISPR/CAS9 Technology”, the authors are suggested to mention that besides indels, DSBs induced by CRISPR-Cas9 can lead to larger genomic rearrangements including large chromosomal deletions, inversions or translocations (PMID: 34365511), as well as plasmid and retrotransposon insertions (PMID: 32095517 and 35760782).
22. Besides Cas9 nuclease, CRISPRi and CRISPRa, Base editors and prime editors should also be mentioned, and are there any Base editor-based paper in GC filed? If yes, it will be good to be added
33. The section of “Application of CRISPR/CAS9 in GC Clinical Research” are suggested to be expanded
Reviewer 2 Report
The authors summarize the current knowledge on the principles of CRISPR/Cas9-based genome editing for basic research and therapeutic intervention in gastric cancer. They focus on the general mechanism of the CRISPR/Cas9 system to modify the genome and a large number of studies using this technology to decipher the pathology of gastric cancer.
In my opinion, the results of the basic research studies are presented rather isolated from each other. The manuscript would therefore greatly benefit from connecting the results of the listed studies to each other in order to better understand the overall impact of CRISPR/Cas9 on basic gastric cancer research. Along this line, a figure depicting the relevant signaling pathways and responsible genes targeted by CRISPR/Cas9 in the studies discussed in the manuscript may be worth considering.
The authors may also want to consider changing the title of the manuscript. Only a marginal proportion of the manuscript deals with the clinical application of CRISPR/Cas9 and I would therefore consider to remove this aspect from the title. The authors may decide whether this is appropriate.
I also believe that the manuscript would benefit from better explaining the relevance of each section at the very beginning. For instance, the first sentence of section 3.2 “Cell cycle Control” reads “The study of genes that influence the cell cycle is important to the identification of specific biomarkers and effective therapeutic targets in GC.” Instead, the authors could emphasize the importance of cell cycle genes for enabling proliferation in gastric cancer which may in turn represent targets for therapeutic intervention.
The authors may also want to consider modifying or even removing Figure 1. It depicts the history of CRISPR and seems only of minor relevance for the remaining content of the manuscript. If they decide to keep the figure, I strongly suggest to revise the content as detailed in the specific points below.
I also want to encourage the authors to change some of the references citing the history and basic mechanism of CRISPR/Cas9 as detailed in the specific points below.
Specific points:
Line 13-14: I would suggest to remove the explanation of the well-known term CRISPR/Cas9 from the abstract and rather explain this at the first use in the introduction. However, please use the correct term “clustered regularly interspaced short palindromic repeats/CRISPR-associated 9” instead of “protein 9 associated with regularly interspaced grouped short palindromic repeats”.
Line 45: Change wording of “last decade” as ZFN have been used for gene editing for >25 years (https://doi.org/10.1073/pnas.93.3.1156).
Line 47: Consider citing the original papers describing ZFN and TALEN instead of review articles which are to my knowledge for ZFN: https://doi.org/10.1073/pnas.93.3.1156 and for TALEN: https://doi.org/10.1534/genetics.110.120717 and https://doi.org/10.1038/nbt.1755.
Line 61: Please also add the major reference demonstrating CRISPR/Cas as adaptive immune system in bacteria which is also described in Figure 1 but not included in the references, yet (https://doi.org/10.1126/science.1138140).
Line 63: The authors correctly mention the discovery of CRISPR/Cas9 as a programmable nuclease in 2012 but cite the wrong reference (Wang et al. 2014). The correct reference for the discovery by Doudna and Charpentier is https://doi.org/10.1126/science.1225829. Shortly afterwards in 2013 they and two other groups adapted the system for use as a gene editing tool (i.e., reviewed here https://doi.org/10.1177/0023677221994613). In this regard the cited reference 17 is not ideal as published in 2014. It is also not consistent with Figure 1 in which Cong et al. is cited for that purpose.
Line 64: The term sgRNA refers to the fused “single-guide RNA”. Instead, I would suggest to either use the general term gRNA for guide RNA or explain the difference to the sgRNA.
Line 66: Capital “Crick” instead of “crick”
Line 71: Although NAG is also recognized to some extend by Cas9, the canonical PAM is defined as only NGG. I would therefore suggest to remove the NAG.
Line 74: “non-homologous end joining” instead of “junction of non-homologous ends” for NHEJ as also depicted like that in Figure 2.
Figure 1: If the authors decide to keep the figure, please ensure that all references are included in the bibliography (at least Barrangou et al. 2007 and Cong et al. 2013 are missing). The authors may also want to revise the wording in some cases. For instance, the first description reads “Ishino et al., Follow-up of the Escherichia coli bacterium” but should emphasize the first description of CRISPR. Cong et al. was published in 2013 and not as indicated in 2012. In addition, CRISPR as a programmable nuclease has been discovered in the Charpentier /Doudna publication in 2012. In addition, is there a reason why the time line stops already at 2014? I would at least include the Nobel prize in 2020 as mentioned in the manuscript (line 50).
Line 91: To my knowledge, the correct reference for the discovery of dCas9 is https://doi.org/10.1016/j.cell.2013.02.022 instead of #21 in the current manuscript.
Line 128: The authors do not discuss the limitations of CRISPR. I would therefore suggest to remove the “despite its limitations” in this paragraph.
Line 269: The advantage of CRISPR over other genome editing tools like ZNFs and TALENs is not versatility but ease of use. The authors may want to change the wording accordingly or elaborate on their statement about the higher versatility of CRISPR.
Line 271: The authors state that CRISPR is challenging in the field of GC but do not explain why.
Line 288: I am unaware of “point” genes. Could the authors please elaborate on that or change the wording?
I would encourage the authors to revise their manuscript, which will subsequently be of interest to the scientific community and will therefore be worth publishing.
